# BAM! Just Like That: Simple and Efficient Parameter Upcycling for Mixture of Experts

**Qizhen Zhang**[1†]    **Nikolas Gritsch**[2,3]    **Dwaraknath Gnaneshwar**[2]    **Simon Guo**[4†]
**David Cairuz**[2]    **Bharat Venkitesh**[2]    **Jakob Foerster**[1]    **Phil Blunsom**[1,2]
**Sebastian Ruder**[2]    **Ahmet Üstün**[3*]    **Acyr Locatelli**[2*]
[1] University of Oxford    [2] Cohere    [3] Cohere For AI    [4] Stanford University

## Abstract

The Mixture of Experts (MoE) framework has become a popular architecture for large language models due to its superior performance over dense models. However, training MoEs from scratch in a large-scale regime is prohibitively expensive. Existing methods mitigate this by pre-training multiple dense expert models independently and using them to initialize an MoE. This is done by using experts' feed-forward network (FFN) to initialize the MoE's experts while merging other parameters. However, this method limits the reuse of dense model parameters to only the FFN layers, thereby constraining the advantages when "upcycling" these models into MoEs. We propose BAM (Branch-Attend-Mix), a simple yet effective method that addresses this shortcoming. BAM makes full use of specialized dense models by not only using their FFN to initialize the MoE layers but also leveraging experts' attention parameters fully by initializing them into a soft-variant of Mixture of Attention (MoA) layers. We explore two methods for upcycling attention parameters: 1) initializing separate attention experts from dense models including all attention parameters for the best model performance; and 2) sharing key and value parameters across all experts to facilitate for better inference efficiency. To further improve efficiency, we adopt a parallel attention transformer architecture to MoEs, which allows the attention experts and FFN experts to be computed concurrently. Our experiments on seed models ranging from 590 million to 2 billion parameters demonstrate that BAM surpasses baselines in both perplexity and downstream task performance, within the same computational and data constraints.

## 1 Introduction

Large language models (LLMs) have demonstrated remarkable performance across a wide range of tasks [2–4]. Often, LLMs use a dense architecture, where models apply all parameters to every input during training and inference [2, 3]. Consequently, increasing model capacity results in increased computational cost. An empirical finding in the scaling laws of LLMs [5] indicates that model performance scales with model capacity. However, naively increasing model size to achieve predicted improvements is challenging due to 1) increased hardware requirements and associated costs; and 2) greater training instabilities such as gradient spikes during large-scale training runs [6]. Mixture of Experts [MoEs; 7–9] have emerged as a popular solution to these issues [10–12]. During training and inference, each input activates only a subset of the model parameters (often referred to as experts), thereby decoupling computation cost from the total parameter count, allowing the total number of

---

*Equal Mentorship

†Work done at Cohere

Corresponding authors: qizhen.zhang@eng.ox.ac.uk, {ahmet, acyr}@cohere.ai

38th Conference on Neural Information Processing Systems (NeurIPS 2024).

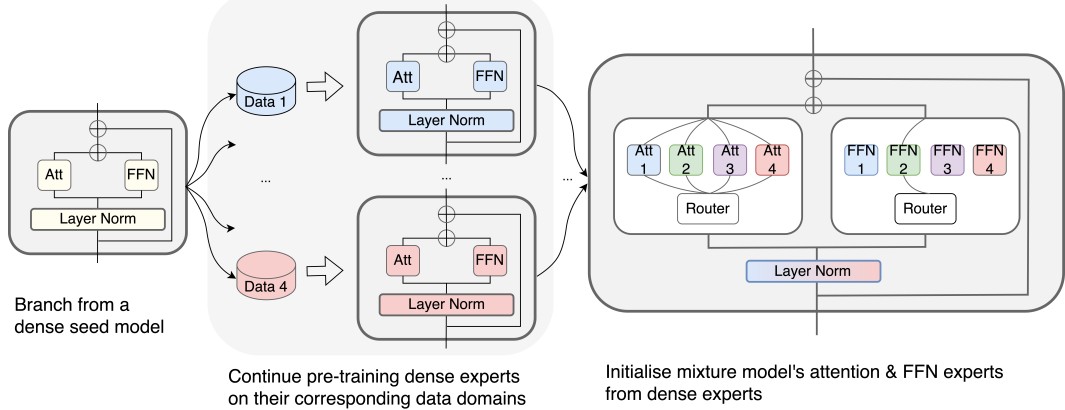

Figure 1: BAM operates in three phases. Different colors correspond to different expert domains, which indicates the pre-trained seed model. White indicates random parameter initialization, gradient color indicates parameter merging **1) Branching**: Begin with a pre-trained dense seed model and create $N$ copies of it. **2) Continued Pre-training**: Continue to pre-train each copy independently on its own data mixture. This process yields specialized dense expert models. **3) Mixture Model Training**: Utilize these specialized dense expert models to initialize both the FFN and attention experts of the mixture model. The router layers are initialized randomly. All other parameters are derived by averaging the corresponding layers in each of the dense experts. Note that BAM employs a parallel attention transformer architecture that concurrently computes attention experts and FFN experts. The figure is loosely based on Figure 1 from Sukhbaatar et al. [1].

parameters to grow without additional computation. Moreover, MoEs empirically outperform dense models with equivalent computational requirements [7, 13].

However, training MoEs from scratch is expensive due larger compute requirements [12, 7] and is difficult due to training instabilities[8]. To address this, recent works have explored more efficient alternatives by initializing MoEs using pre-trained dense models [14, 1], followed by a small number of MoE training steps. This approach offers several advantages: 1) By utilizing a pre-trained model, we can leverage existing knowledge rather than starting from a random initialization; 2) the dense model used for initialization is smaller in total parameter count, making it faster and easier to train; 3) various pre-trained dense models are readily available in the open source community, which can be improved further without incurring additional pre-training cost [15, 16, 3]. In particular, Branch-Train-MiX [BTX; 1] initializes an MoE with $N$ FFN experts through a three-step upcycling process. Initially, $N$ copies of a pre-trained seed model are created. Each copy is then further pre-trained on domain-specific data for specialization. The MoE's FFN expert parameters are subsequently initialized from these specialized dense models by directly copying their FFN parameters. It is important to note that the MoE's non-FFN parameters, such as attention layers, are not converted to MoE layers. Since there are $N$ dense models, each with its own set of attention parameters, BTX uses the average of the $N$ attention parameters to initialize the MoE's attention module while the router parameters are initialized randomly. However, this approach under-utilizes the potential of the specialized dense models for two key reasons. 1) The expert knowledge embedded in the attention modules of each of these specialized models is not leveraged, despite their potential contribution to performance enhancement; 2) Uniform averaging of parameters from distinct models, even though they originate from the same seed, can compromise performance due to averaging divergent specializations [17].

Motivated by these limitations, we propose Branch-Attend-Mix (BAM), a simple yet effective improvement to BTX that upcycles both FFN and attention parameters into expert layers. See Figure 1 for an illustration of BAM. Our contributions are as follows:

- To fully leverage the benefits of specialized attention experts and improve training stability, BAM uses a variant of Mixture of Attention [MoA; 18] with soft routing where each token is assigned to every attention expert. We validate through ablation studies that this is critical to surpass baseline performance.

- We propose two variants of BAM: 1) For best model performance, BAM with KV Experts includes all attention parameters in the attention experts; 2) For enhanced inference efficiency, BAM with KV sharing shares the key and value parameters across all attention experts, similar to the method done in Zhang et al..

- To alleviate the increased computations from soft-routing attention experts, BAM employs a parallel attention transformer architecture [19, 20] that allows the attention experts and FFN experts to be computed in parallel, thereby improving throughput.

- We validate our approach through experiments on seed models ranging from 590 million to 2 billion parameters. Our results demonstrate that BAM outperforms the BTX model in terms of both perplexity and downstream task performance across various domains under equivalent data and computational constraints, confirming the effectiveness of our proposed enhancements.

## 2 Related Work

**Efficient initialization for MoEs**  Two recent works have studied how to use pre-trained dense models to initialize MoEs [14, 1]. Sparse Upcycling [14] upcycles a single dense model by simply copying over dense model parameters as MoE parameters. It replicates FFN parameters $N$ times into each FFN expert in sparse MoE layers. BTX [1] improves on this approach by upcycling not just a single dense seed model but also multiple specialized dense models branched from the seed model. This is advantageous because not only can the dense models be trained in parallel, but also they are specialized in particular domains. However, unlike our method, both approaches only upcycle the FFN part of the dense (seed or specialized) models.

**Alternative architecture for MoEs**  Alternative to the standard MoE architectures where only feed-forward blocks (FFNs) are used as MoE layers, recent work [18, 21–23] investigates the mixture-of-attention framework, which uses attention experts in addition to FFN experts. This approach has not gained widespread popularity because it only achieves modest performance improvements while requiring additional engineering tricks for optimization [22]. However, in the setting of upcycling specialized dense experts into an MoE, the usage of attention experts is much more appropriate, as the attention parameters contain specialized domain knowledge that would otherwise not be accessible in the final MoE.

**Alternatives Methods for Model Merging**  Recent work has tried to combine many openly available dense models to create an improved model, via layerwise merging or stitching of pre-trained model weights [24]. However, this approach creates a new dense model, not a sparse one. Other work has looked into dynamically adding new blocks to MoA whenever the model is trained on a new domain, but uses random initialization for the new experts instead of utilizing existing models [25].

## 3 Background

### 3.1 Parallel Attention Transformers

The typical architecture for LLMs is based on stacking multiple blocks of the transformers [26]. The conventional transformer block comprises a Multi-Headed Attention (MHA) module, commonly referred to as the attention layer, followed by a residual connection and a feed-forward neural network (FFN). Recent works [19] introduced the parallel attention transformer, also known as parallel layers [20]. Parallel attention transformers is a variant of the original architecture which computes the outputs of the attention and the FFN layers in parallel using the same input. The final transformer output is the projected concatenation of the attention output and the FFN output with its residual connection (See Figure 1). Processing the two layers in parallel increases the computational throughput without degrading the performance [20].

### 3.2 Multi-Headed Attention

In an attention layer with $h$ heads, the input is linearly projected to form the query, key, and value vectors through learned parameters. Each head applies the attention mechanism to its vectors independently to compute its attention outputs. The individual outputs from all $h$ heads are concatenated

and linearly projected back to the dimensionality of the original model, ensuring consistent input and output dimensions throughout the network layers.

$$\text{Head}_i = \text{softmax}\left(\frac{QW_i^q(KW_i^k)^T}{\sqrt{d_k}}\right)VW_i^v$$

$$\text{MHA}(Q, K, V) = \text{concatenate}\left(\text{head}_1, \ldots, \text{head}_h\right)W^o, \tag{1}$$

where $W_i^q, W_i^k \in \mathbb{R}^{d_{\text{model}} \times d_k}$, $W_i^v \in \mathbb{R}^{d_{\text{model}} \times d_v}$ and $W^o \in \mathbb{R}^{hd_v \times d_{\text{model}}}$, $d_k$ is the key dimension.

### 3.3 Mixture of Experts

The Mixture of Experts [MoE; 9] model replaces the FFN layer by an MoE layer in the conventional dense transformer. An MoE layer consists of a linear router, parameterized by $W_{\text{FFN router}} \in \mathbb{R}^{d_{\text{model}} \times N}$, and a set of $N$ FFN experts, denoted as $\{\text{FFN}_i(x)\}_{i=1}^N$. The router outputs the normalized router logits $p(x) = \text{softmax}\left(W_{\text{FFN router}} \cdot x\right)$, where $p_i(x)$ corresponds to the gate value for the $i$-th expert, $\text{FFN}_i$. The router assigns the input token representation $x$ to the $k$ experts with the highest gate values. Denoting the set of selected top-$k$ indices as $\mathcal{T}$, the MoE layer's final output is the weighted sum of the selected experts' outputs, weighted by their respective gating values:

$$\text{FFN}_{\text{MoE}} = \sum_{i \in \mathcal{T}} p_i(x)\text{FFN}_i(x).$$

### 3.4 Mixture of Attention

Mixture of Attention [MoA; 18, 23] extends MoEs by also replacing the conventional attention layer with attention experts. Each MoA layer comprises of a linear router parameterized by $W_{\text{attn router}} \in \mathbb{R}^{d_{\text{model}} \times N}$ and a set of $N$ attention experts, $\{\text{MHA}_j(x)\}_{j=1}^N$. The router outputs the normalized router logits $g(x) = \text{softmax}\left(W_{\text{attn router}} \cdot x\right)$, with each $g_j(x)$ represents the gate value for the $j$-th expert. Each attention expert is composed of query-projection experts $\{W_j^q\}_{j=1}^N$ and the attention output projection experts $\{W_j^o\}_{j=1}^N$. The key projection $W^k$ and value projection $W^v$ are shared among all experts for computation efficiency. The final output of the MoA layer is a gate-value weighted sum of the computations from the top-$k$ selected experts $\mathcal{M}$:

$$\text{MHA}_{\text{MoA}} = \sum_{i \in \mathcal{M}} g_i(x)\text{MHA}_i(x).$$

## 4 BAM: Branch-Attend-Mix

There are three phases to BAM:

**1) Branching**: We create $N$ copies of the model from a pre-trained dense model. We will refer to this dense model as the seed model from here on. We use a pre-trained seed model with a parallel attention transformer architecture. This is motivated by two reasons: 1) Parallel attention transformers offer better throughput compared to traditional transformers, with only minor performance degradation at smaller scales but no degradation at larger scales [20]. 2) Employing a dense seed model with a parallel attention transformer architecture allows the upcycled mixture model to inherit this parallel attention setup. This architecture enables the attention experts and FFN experts to be computed concurrently, thereby enhancing both training and inference efficiency.

**2) Continued Pre-training**: Each copy of the seed model undergoes independent pre-training on a specialized data mixture, each tailored to different domains such as law, mathematics, and coding. This phase results in specialized dense expert models that have enhanced performance within their respective domains compared to the seed model. However, these models might perform wose in domains outside their specialization.

**3) Mixture Model Training**: We initialize the mixture model using the specialized dense models from step 2. We also incorporate a copy of the original pre-trained dense seed model (similar to the BTX approach), ensuring the seed model's general knowledge acquired in the pre-training stage

is transferred to the mixture model. A key differentiator for BAM is it takes full advantage of the dense specialized models by leveraging both the FFN and attention layers. The parameters for each attention expert and FFN expert in the mixture model are directly inherited from their corresponding expert or seed dense model. Non-expert parameters in the mixture model, such as layer normalization, input and output embeddings, are initialized by uniformly averaging the corresponding parameters from the dense models. The router layers are randomly initialized. The upcycling of both FFN experts and attention experts is beneficial for two reasons: 1) It maximizes the utilization of specialized capabilities in the dense models; and 2) It avoids averaging the attention parameters, which has shown to degrade model performance [17]. In the subsequent section, we will discuss in more detail the training methodologies of the mixture model.

## 4.1  Mixture Model Training: Attention Experts

The MoA architecture proposed by Zhang et al. uses only the query projection ($W^q$) and the attention output projection ($W^o$) as attention experts, while sharing the key-value projections among all experts to enhance efficiency. Like conventional MoEs, Zhang et al. also employs top-k routing, which assigns input tokens to only $k$ experts with the highest attention gate values.

To adapt MoA to our BAM framework, we propose two modifications to the original architecture. First, instead of the traditional top-$k$ routing, BAM employs soft routing for the attention experts, where the attention router assigns each input token to all attention experts. We show in Section 7 that using soft-routing is crucial for achieving performance superior to the baseline. Second, we propose a variant of MoA where that also incorporates the key-value projections ($W^k$ and $W^v$) as part of the attention experts (i.e. we have one copy of $W_i^k, W_i^v$ for each attention expert $i$). This is as oppose to than sharing them across all experts. These two modifications are motivated by three reasons:

1. Employing soft-routing and including KV experts allow us to fully exploit the capabilities of the specialized attention experts.

2. In transformer architectures, FFNs computations typically account for a larger portion of computational load than attention computations [27, 28]. By adopting the parallel attention transformer architecture, BAM enables the concurrent computation of both the attention experts module and the FFN experts module. In addition, we implement expert parallelism [7] where the individual FFNs and attention are computed in parallel across experts. Thus, we are partially "masking" the extra computations introduced by soft-routing and KV experts under the more expensive FFN experts computations.

3. Using soft-routing allows for more stable training dynamics since the router is no longer a discrete optimization problem. This alleviates the attention experts training from common MoE problems such as imbalanced router load[7].

We discuss in Section 5.3 on how to evaluate BAM fairly comparing to baselines.

## 4.2  Mixture Model Training Phase: FFN Experts

We use top-1 routing of FFN experts for all experiments unless otherwise stated. For training stability, we use the auxiliary load balancing loss $\mathcal{L}_{\text{LB}}$ [7] to encourage uniform load across all experts and the auxilliary router z-loss [8] to penalize large logits going into the router. Let $B$ be the number of tokens in the batch, $f_i$ be the fraction of tokens assigned to FFN expert $i$, and $P_i$ be the fraction of the expert $i$'s gate value, then the two auxiliary losses are as follows:

$$
\begin{aligned}
\mathcal{L}_{\text{LB}} &= N \sum_{i=1}^{N} f_i P_i \\
\mathcal{L}_z &= \frac{1}{B} \sum_{i=1}^{B} \left( \text{LogSumExp}_{j=1:N} \left( x_j^i \right) \right)^2
\end{aligned}
\tag{2}
$$

The model's final loss function $\mathcal{L} = \mathcal{L}_{\text{NLL}} + \sum_{\forall \text{ MoE Layers}} \left( \alpha \mathcal{L}_{\text{LB}} + \beta \mathcal{L}_z \right)$ is a weighted sum, where $\mathcal{L}_{\text{NLL}}$ is the negative log-likelihood loss, and $\alpha$ and $\beta$ are hyperparameters.

# 5 Experiment Details

## 5.1 Experimental Setup

We validate BAM against the baseline model, BTX, through two sets of experiments that vary in scale. Detailed architecture specifications are shown in Table 8 in the Appendix. For both sets of experiments, we train a mixture model upcycled from four dense experts: a general pre-training seed model, along with specialized experts in code, mathematics, and law. Each expert undergoes a continued pre-training phase using 100 billion tokens. We use the AdamW optimizer [29] with a weight decay of $0.1$. To ensure training stability, the peak learning rate for the continued pre-training phase is set to 50% that of the pre-training phase.

**Small-Scale Experiments** Our first set of experiments uses a dense seed model with 590 million parameters, pre-trained on 400 billion token with a batch size of 1 million tokens. During the pre-training and mixture model training phases, we employ a learning rate schedule that warms up from 0 to $1.2e-4$ over 1500 steps, then undergoes a cosine decay to 10% of the peak learning rate.

**Large-Scale Experiments** The second set of experiments utilizes a dense seed model with 2 billion parameters, pre-trained on 750 billion tokens with a batch size of 4 million tokens. During the pre-training phase, the learning rate schedule warms up from 0 to $1e-2$ over 2000 steps, then cosine-decays to 3% of the peak learning rate. In the mixture model training phase, we observes gradient spikes when using the same learning rate schedule as in the pre-training phase. To mitigate this, we lower the peak learning rate to $1e-4$, warm-up steps to 1000. And as before, cosine-decay to 10% of the peak learning rate.

## 5.2 Data Details

Below we describe the data details for both BAM and BTX experiments. In the **continued pre-training phase**, each dense expert is trained on a specialized data domain for 100 billion tokens. The data domains contain the following data:

- **Mathematics**: Utilizes the MathGLM [30], GSM8K [31], proof-pile-2 [15], and MathPile [32] datasets. Notably, we ensure that there is no overlap between the GSM8K train and evaluation splits.
- **Code**: Comprises the Starcoder dataset [33].
- **Law**: Includes the pile-of-law [34] and HUPD [35] datasets.

Additionally, each data domain is augmented with 10% data sourced from Common Crawl. In our ablation experiments, we observed that incorporating a portion of general-text data into the expert training mix enhances model performance performance as it allows the expert data distribution to more matching with the data distribution of the seed model.

In the **mixture model training phase**, the training data comprises an equal distribution of data from each domain (including the pre-training domain), with each domain contributing 25% to the overall mix.

## 5.3 Evaluation

We compare BAM to BTX, as well as against the original dense seed model and the dense expert models. Similar to the evaluations in Sukhbaatar et al. [1], BAM is evaluated in two settings:

- **Data-Matching (DM)**: We use exactly the same training data in terms of both content and quantity for BAM as used for the BTX baseline.
- **Compute-Matching (CM)**: We train BAM with the same amount of computation resources as BTX is trained on, measured in TPU-core-days. In the small-scale experiments, all MoE training phases are allocated 25 TPU-core-days, while in the large-scale experiments, the allocation increases to 305 TPU-core-days.

|  | Pretrain | Code | Law | Math | Average |
|---|---|---|---|---|---|
| Base Model | **23.98** | 8.62 | 8.92 | 14.12 | 13.91 |
| *Specialized Models* | | | | | |
| Code Dense Expert | 39.50 | 3.71 | 10.83 | 8.77 | 15.70 |
| Law Dense Expert | 87.82 | 23.45 | 7.53 | 30.79 | 37.40 |
| Math Dense Expert | 57.61 | 6.78 | 10.43 | 5.67 | 20.12 |
| *Generalist Models* | | | | | |
| BTX (Baseline) | 26.72 | 3.78 | 6.63 | 5.77 | 10.72 |
| BAM DM (Expert KV), ours | 24.02 | **3.45** | **6.04** | **5.31** | **9.70** |
| BAM CM (Expert KV), our | 24.84 | 3.53 | 6.21 | 5.43 | 10.00 |
| BAM DM (Shared KV), ours | 25.67 | 3.64 | 6.39 | 5.57 | 10.32 |
| BAM CM (Shared KV), ours | 26.00 | 3.72 | 6.50 | 5.63 | 10.46 |

Table 1: Perplexity evaluation (↓) of BAM versus BTX for small-scale experiments (using a seed model with **590M parameters**). Highlighted entries denote models that outperform the BTX baseline. BAM consistently outperforms the baseline under both compute matching and token matching regimes.

|  | Pretrain | Code | Law | Math | Average |
|---|---|---|---|---|---|
| Base Model | **9.83** | 2.41 | 3.86 | 3.71 | 3.33 |
| *Specialized Models* | | | | | |
| Code Dense Expert | 15.39 | **2.18** | 5.22 | 4.34 | 3.91 |
| Law Dense Expert | 32.69 | 6.84 | **3.09** | 8.61 | 6.18 |
| Math Dense Expert | 20.32 | 3.20 | 5.11 | **3.20** | 3.84 |
| *Generalist Models* | | | | | |
| BTX (Baseline) | 10.35 | 2.40 | 3.76 | 3.64 | 3.27 |
| BAM DM (Expert KV), ours | 10.11 | 2.36 | 3.66 | 3.55 | **3.19** |
| BAM CM (Expert KV), our | 10.19 | 2.37 | 3.69 | 3.57 | 3.21 |
| BAM DM (Shared KV), ours | 10.20 | 2.37 | 3.69 | 3.59 | 3.22 |
| BAM CM (Shared KV), ours | 10.28 | 2.38 | 3.72 | 3.61 | 3.24 |

Table 2: Perplexity evaluation (↓) of BAM versus BTX for large-scale experiments (using a seed model of **2B parameters**). Highlighted entries indicate models outperforming the BTX baseline. BAM consistently surpasses BTX in all domains under both compute matching and token matching regimes.

In addition to evaluating perplexity, we also asses large-scale models' performances with zero-shot and few-shot downstream tasks relevant to the expert domains[1].

For each domain, we report the average of task scores within that domain.

- **Math**: For mathematical reasoning, we report the average performance on GSM8K (8-shot) [31] and MATH (4-shot) [36].

- **Code**: For code generation, we evaluate on HumanEval (0-shot) [37] and MBPP (3-shot) [38].

- **Law**: We use the International Citizenship Questions sub-task in LegalBench [39].

- **World Knowledge**: For general knowledge on facts, we report the performance on Natural Questions (0-shot) [40] and TriviaQA (0-shot) [41].

- **Reasoning**: For reasoning abilities, we use ARC-Easy, ARC-Challenge [42], SIQA [43], PIQA [44], WinoGrande [45], and QUAC [46] (all 0-shot).

- **General Knowledge**: We use MMLU (5-shot) [47] to test general language understanding.

---

[1]For LegalBench, we specifically selected the International Citizenship QA sub-task, as it was the only sub-task among those implemented in our code-base that scored above random.

| | Math | Code | Law | Know. | Reason. | MMLU | Average |
|---|---|---|---|---|---|---|---|
| Seed Model | 3.68% | 9.41% | 73.34% | **21.33%** | **47.73%** | 34.13% | 31.60% |
| *Specialized Models* | | | | | | | |
| Math Dense Expert | **4.92%** | 12.39% | 68.21% | 13.32% | 46.11% | 34.29% | 29.87% |
| Code Dense Expert | 3.19% | **18.80%** | 21.49% | 12.18% | 44.29% | 31.50% | 21.91% |
| Law Dense Expert | 3.05% | 0.20% | **88.80%** | 10.41% | 44.08% | 32.18% | 29.79% |
| *Generalist Models* | | | | | | | |
| BTX (Baseline) | 3.86% | 10.05% | 81.85% | 19.07% | 47.36% | 34.07% | 32.71% |
| BAM DM (Expert KV), ours | 4.44% | 12.83% | 85.47% | 19.89% | 47.11% | 34.42% | **34.02%** |
| BAM CM (Expert KV), ours | 4.34% | 12.48% | 82.79% | 19.51% | 47.43% | 34.43% | 33.50% |
| BAM DM (Shared KV), ours | 4.10% | 11.76% | 86.73% | 19.48% | 47.27% | **34.55%** | 33.98% |
| BAM DM (Shared KV), ours | 3.65% | 11.77% | 80.98% | 19.22% | 47.56% | 34.16% | 32.89% |

Table 3: Benchmark evaluations (↑) for BAM versus BTX. Table shows large-scale experiments (using a seed model of **2B parameters**). Highlighted entries indicate models outperform the BTX baseline. All BAM variants outperform BTX on average.

# 6   Results

We evaluate two variants of BAM: *BAM (Expert KV)*, where all attention parameters are utilized in the attention experts, and *BAM (Shared KV)*, where key-value (KV) projections are shared among all attention experts. We report evaluations on both variants because each presents its own advantages. *BAM (Expert KV)* yields better model performance because it leverages more specialization from expert dense models. Conversely, *BAM (Shared KV)* offers greater inference efficiency because of reduced memory requirements and the shared KV cache. A more detailed analysis on inference efficiency is provided in Section 7.

**Perplexity Evaluations**   We present perplexity results for both dense models (seed and specialized dense models) and generalist models (BAM and BTX). Results from small-scale experiments are shown in Table 1, and large-scale experiments in Table 2. In all experiments, BAM consistent outperforms BTX under both the compute and token-matching regimes. While dense expert models sometimes excel in their respective specialized domain, they significantly under perform compared to generalist MoE models in all other domains.

**Benchmark Evaluations**   See Table 3 for downstream tasks evaluations on large scale experiments where we show that all BAM variants outperform BTX on average. We omit the small-scale downstream results as most of the performance was indistinguishable from random guessing, likely due to the limited scale of the models.

# 7   Ablations

## 7.1   On the Importance of Upcycling Attention Experts

We ablate the importance of upcycling attention experts in BAM by comparing it to BTX, ensuring both models have matching total parameters (equivalent computations) and/or active parameters (equivalent model capacity). All ablation experiment use the same number of 100B tokens in the mixture model training phase. See Table 9 in the Appendix for a detailed walk-through on calculating the number of total and active parameters of each model.

**Matching the Number of Total Parameters with BTX**    To match the number of total parameters in BTX with BAM, we simply increase the number of FFN experts while still using top-1 routing. All newly added FFN experts are initialized from the seed model's FFN. As shown in Table 4, BTX matches BAM's total parameter count by employing 6 FFN experts. Even when BTX's total number of FFN experts is increased to 8, thereby exceeding the total parameter count of BAM, BAM continues to outperform BTX.

|                | Total Params | Active Params | Code | Law  | Math | Pre-train |
|----------------|--------------|---------------|------|------|------|-----------|
| BAM, *ours*    | 776M         | 663M          | **3.44** | **6.04** | **5.31** | **24.06** |
| BTX 4 Experts  | 700M         | 587M          | 3.78 | 6.63 | 5.77 | 26.72     |
| BTX 5 Experts  | 738M         | 587M          | 4.11 | 7.19 | 6.25 | 29.08     |
| BTX 6 Experts  | 776M         | 587M          | 3.99 | 7.03 | 6.10 | 28.50     |
| BTX 7 Experts  | 813M         | 587M          | 3.90 | 6.84 | 6.00 | 27.36     |
| BTX 8 Experts  | 851M         | 587M          | 3.94 | 6.92 | 6.05 | 27.79     |

Table 4: Perplexity (↓) ablation studies for small-scale experiments assess how BAM's total parameter count compares with BTX. To match BTX's total parameters with those of BAM, we incrementally increase the number of FFN experts in BTX from 4 up to a maximum of 8. The additional experts are upcycled using the FFN parameters from the same dense seed model.

**Matching the Number of Total & Active Parameters with BTX**    To match the number of active parameters with BAM, we simply increase the number of top-k from the usual top-1 to top-3 in every MoE layer of BTX. We see that BTX matches total parameters and active parameters with BAM when using top-3 routing with a total of 6 experts in Table 5. 3 of the 6 experts are upcycled from the 3 expert dense models while the other 3 are upcycled from the same copy of the dense seed model. Despite matching parameters, BTX does not outperform BAM.

|                        | Total Params | Active Params | Code | Law  | Math | Pre-train |
|------------------------|--------------|---------------|------|------|------|-----------|
| BAM, *ours*            | 776M         | 662M          | **3.44** | **6.04** | **5.31** | **24.06** |
| BTX top-3 w/ 6 Experts | 776M         | 662M          | 3.55 | 6.23 | 5.48 | 24.13     |

Table 5: Perplexity ablation studies on small-scale experiments compare the performance of BAM to BTX under conditions where both models have equivalent numbers of active parameters and total parameters. In addition to utilizing six FFN experts, we have adjusted BTX's routing mechanism from top-1 to top-3 routing. This change means that instead of activating just one FFN expert per token representation, three are now activated, effectively increasing the number of active parameters per input.

## 7.2   On the Importance of Using Soft-Routing for Attention Experts

We compare soft routing with the most commonly used sparse routing approaches (i.e. top-1 and top-2 routing) in BAM's attention experts layers. See Table 6 for ablation results. All experiments are compute matched at the 590M scale. Our results indicate that BAM with top-1 and top-2 attention experts routing does not show improvement over the baseline BTX on most domains. Conversely, BAM with soft-routing consistently outperforms BTX across all domains. This demonstrates that using soft routing in MoA layers is crucial and supports our decision to implement this approach.

|                        | Pretrain | Code | Law  | Math | Average |
|------------------------|----------|------|------|------|---------|
| BTX                    | 26.72    | 3.78 | 6.63 | 5.77 | 10.72   |
| BAM soft-routing MoA   | **26.00** | **3.72** | **6.51** | **5.64** | **10.47** |
| BAM top-2 routing MoA  | 26.68    | 3.78 | 6.69 | 5.75 | 10.72   |
| BAM top-1 routing MoA  | 26.89    | 3.83 | 6.69 | 5.82 | 10.81   |

Table 6: Perplexity ablation (↓) of BAM compute matched with different attention experts routing methods.

## 7.3   Analysis on Inference Efficiency

We analyze the FLOPs count per token [5, 48] for BTX versus BAM during inference. In Table 7, we provide the estimates for non-embedding parameter per Transformer layer, excluding non-linearities,

biases, and layer normalization as they are negligible. Our results are given for the small-scale model dimensions stated in Table 9 in the Appendix. For the detailed FLOPs calculation steps, refer to Table 10 in the Appendix. Compared to the standard BTX, BAM consumes more FLOPs due to the soft-routing attention experts. However, the increase in FLOPs is partially mitigated by our implementation of a parallel attention transformer architecture and expert parallelism. In addition, when we compare FLOPs between BAM and the parameter-matching variant of BTX, BAM contains approximately the same number of FLOPS while still performing better model-quality wise (see Table 5).

For empirical inference latency, we measure the time taken to generate 16 tokens from a prompt of length 256, averaged over 10 runs on TPUs. We observe that BAM is slightly slower (6.17s) than standard BTX (4.81s). The increased latency in BAM is likely attributable to increased memory pressure from the individual KV caches maintained by each attention expert, as attention mechanisms are typically memory-bound [49]. Motivated by this, the KV-sharing version of BAM reduces the inference latency from 6.17s to 5.96s, as all attention experts share a single KV cache. In future work, can further optimize inference time to be closer to the standard BTX baseline (representative of standard MoEs) as our hardware and framework code is optimized for training rather than inference latency.

| Method | Config | | Params | | FLOPs | | |
| --- | --- | --- | --- | --- | --- | --- | --- |
| | $n_{experts}$ | $n_{topk}$ | Total | Active | Attention | FFN | Total |
| BAM | 4 | 1 | 776M | 663M | 35,651,584 | 12,589,056 | 48,257,024 |
| BTX | 4 | 1 | 700M | 587M | 8,912,896 | 12,589,056 | 21,510,144 |
| BTX | 6 | 3 | 776M | 663M | 8,912,896 | 37,767,168 | 46,692,352 |

Table 7: Estimates for FLOPs per token during inference. Row 1 shows BAM with KV experts, row 2 shows the standard BTX, and row 3 shows a parameter-matching variant of BTX with 6 FFN experts & top-3 routing (refer to Table 5 in the paper). We use a prompt with context length of 256.

## 8 Conclusion & Future Work

In this work, we introduce BAM (Branch-Attend-Mix), a simple yet effective improvement for upcycling dense models into MoE models. In addition to upcycling just the FFN parameters, we also upcycle the dense model's attention parameters into attention experts. To maximize the benefits of specialized dense models and enhance training stability, BAM uses a variant of Mixture of Attention with soft routing whereby each token is assigned to every attention expert. Our ablation studies confirm that this approach is crucial for surpassing baseline performance levels. We propose two variants of BAM to address different needs: BAM with KV Experts, which includes all attention parameters in the attention experts for optimal model performance, and BAM with KV Sharing, which shares the key and value parameters across the attention experts to boost inference efficiency. To alleviate the increased computations from soft-routing attention experts, BAM employs a parallel attention transformer architecture. This setup enables concurrent computation of the attention expert sand FFN experts. Finally, we validate our approach through experiments on seed models ranging from 590 million to 2 billion parameters. Our results consistently show that BAM outperforms the BTX model in both perplexity and downstream task performance across various domains. These outcomes validate the effectiveness of the enhancements proposed in BAM. Future work could focus on optimizing the training data mixture for BAM's three phases and improving the training framework to accelerate both training and inference processes.

## Acknowledgements

The authors would like to thank Siddhartha Rao Kamalakara for his helpful pointers on debugging the inference code. The authors thank Felipe Cruz for his tremendous help in training and evaluating the seed model used in the large scale experiments. The authors also thank Amr Kayid, Cécile Robert-Michon, Joon Kim, Milad Alizadehand, Louise Rust, and Autumn Moulder for their amazing support on GPU and TPU infrastructure. The authors are grateful to John Lin, Viraat Aryabumi, and Roman Castagné for providing valuable suggestions and insights on data and evaluation. QZ also thanks Joanna Yoo, Kuba Perlin, and Sid for their help and discussions on a previous related project on MoEs sometime ago. QZ thanks Chris Lu for his discussions and feedback on the project.

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

# A  Model Architecture Details

| | Large Scale | Small Scale |
|---|---|---|
| Embedding dimension | 2304 | 1024 |
| FFN dimension | 18432 | 4096 |
| # Heads | 18 | 8 |
| # KV heads | 6 | 8 |
| Vocabulary Size | 256000 | 256000 |
| Activation Function | swiglu | swiglu |
| # layers | 18 | 6 |
| Positional Embedding | rope | rope |
| Share Input & Output Embedding | Yes | No |
| Seed Model Parameters | 2 Billion | 590 Million |

Table 8: Model architecture details for large and small scale experiments

.

| | Dense | BAM | BAM (KV sharing) | BTX top-1 | BTX top-3 |
|---|---|---|---|---|---|
| layernorm / Block | 1024 | 1024 | 1024 | 1024 | 1024 |
| attn_out / Block | 1,048,576 | 4,194,304 | 4,194,304 | 1,048,576 | 1,048,576 |
| qkv_proj / Block | 3,145,728 | 1,258,2912 | 4,194,304 | 3,145,728 | 3,145,728 |
| ffn_exp / Block | 4,194,304 | 4,194,304 | 4,194,304 | 4,194,304 | 12,582,912 |
| ffn_red / Block | 2,097,152 | 2,097,152 | 2,097,152 | 2,097,152 | 6,291,456 |
| router / Block | 0 | 4096 | 4096 | 4096 | 4096 |
| input embedding | 262,144,000 | 262,144,000 | 262,144,000 | 262,144,000 | 262,144,000 |
| output embedding | 262,144,000 | 262,144,000 | 262,144,000 | 262,144,000 | 262,144,000 |
| layernorm | 1024 | 1024 | 1024 | 1024 | 1024 |
| **Active Params** | 587,209,728 | 662,731,776 | 612,400,128 | 587,234,304 | 662731776 |
| **Total Params** | 587,209,728 | 776,002,560 | 738,253,824 | 700,480,512 | 700,480,512 |

Table 9: Active and total parameter counts for small scale ablation experiments. All transformer parameters are reported as per transformer block.

# B Inference Efficiency Analysis

| | Ops | | BTX | BAM |
|---|---|---|---|---|
| **Attention** | Attention Router | Params | - | $n_{experts}d_{model}$ |
| | | FLOPs | - | $2n_{experts}d_{model}$ |
| | Attention: QKV | Params | $d_{model}3d_{attn}$ | $n_{experts}d_{model}3d_{attn}$ |
| | | FLOPs | $6d_{model}^2$ | $6n_{experts}d_{model}^2$ |
| | Attention: Mask | Params | - | - |
| | | FLOPs | $2n_{ctx}d_{model}$ | $2n_{experts}n_{ctx}d_{model}$ |
| | Attention: Projection | Params | $d_{attn}d_{model}$ | $n_{experts}d_{attn}d_{model}$ |
| | | FLOPs | $2d_{model}^2$ | $2n_{experts}d_{model}^2$ |
| **FFN** | FFN Router | Params | $n_{experts}d_{model}$ | $n_{experts}d_{model}$ |
| | | FLOPs | $2n_{experts}d_{model}$ | $2n_{experts}d_{model}$ |
| | FFN | Params | $1.5n_{experts}d_{model}d_{ff}$ | $1.5n_{experts}d_{model}d_{ff}$ |
| | | FLOPs | $3n_{topk}d_{model}d_{ff}$ | $3n_{topk}d_{model}d_{ff}$ |

Table 10: Detailed breakdown of parameters and FLOPs per token in forward pass (inference) for each MoE layer, comparing the standard BTX and BAM with KV experts. Note that BAM uses soft-gating MoA which utilizes all the attention experts, increasing total attention FLOPs and introduces a second router / gating network for attention experts.

