# OpenReview forum: "BAM! Just Like That: Simple and Efficient Parameter Upcycling for Mixture of Experts"
_NeurIPS.cc/2024/Conference — NeurIPS 2024 poster_

### Official Review · Reviewer_ScVn · 2024-06-14

**Soundness:** 1
**Presentation:** 2
**Contribution:** 2
**Rating:** 6
**Confidence:** 5

**Summary:**

This paper proposes a method for upcycling specialized dense models into mixture models. The upcycling is applied to both FFNN and Attention, within a parallel-attention transformer architecture. The evaluation shows their method’s superiority upon a baseline that only upcycles the FFNN layers.

**Strengths:**

* The paper is well put together with clear figures and illustrations of methodology.
* The improvement by applying MoA upon baseline (i.e., BTX) is well-motivated.

**Weaknesses:**

* Table 3 shows the superiority on average of BAM over BTX. However, the specialized models seem not that specialized in their corresponding tasks, which makes the conclusion less convincing. For example, the “Law Dense Expert” beats the “Math Dense Expert” on the mask task.
* This work employs a parallel-attention transformer architecture which is not quite popular. Since the authors state it “increases the computational throughput without degrading the performance”, it would be better to conduct more ablation experiments to further understand its influence on performance and efficiency.
* The MoA requires “additional engineering tricks for optimization” as the authors said, it would be better to conduct further ablations studies by comparing the performance with BTX under the same inference throughput budget.
* It would be better to understand the role of soft routing, instead of vanilla sparse routing, in the MoA layers either from empirical or theoretical perspectives.

**Questions:**

See above.

**Limitations:**

Yes.

---

> ### Author Rebuttal · Authors · 2024-08-07
>
> Dear reviewer,
>
>
> Thank you for your valuable feedback! We will recap your concerns/questions and address them one by one as follows.
>
> > Concern #1: Specialized models seem not that specialized in their corresponding tasks, making the conclusion less convincing.
>
> Thank you for highlighting this issue. We observe this unspecialized behavior in two areas of our original submission’s results:
>
> 1) **In the 590M models' downstream evaluations**: We recommend interpreting the abilities of the 590M models primarily through the perplexity results and disregarding the downstream tasks results. The performance of these models on downstream tasks is very noisy and models are close to random guessing in most tasks due to their small size. We initially included these results for readers who might be curious about downstream evaluations at this scale, but for clarity, we will remove them from the camera-ready copy.
> 2) **In the 2B models' downstream evaluations**: Specifically, the "Math Dense Expert" outperforms the "Law Dense Expert" on the law task. Upon further examination, we found that the LSAT benchmark [1] we used for evaluating law downstream performance primarily tests general reasoning abilities [1] rather than legal knowledge. Consequently, we re-evaluated all models using subtasks from LegalBench [2], an appropriate benchmark for law tasks. As shown in Table 2 of the updated PDF (see global response), all dense experts now perform best within their respective domains. Our conclusion that BAM outperforms baselines remains robust.
>
> TLDR: By excluding the noisy downstream evaluations of small-scale experiments, our results indeed show that specialized models do specialize in their corresponding tasks. Our conclusion holds true through these results.
>
>
> [1] From LSAT: The Progress and Challenges of Complex Reasoning, 2021
>
>
> [2] LegalBench: A Collaboratively Built Benchmark for Measuring Legal Reasoning in Large Language Models, 2023
>
>
> > Concern #2: Parallel-attention transformer architecture is not quite popular […]
>
> Both PALM [3] (5000+ citations since 2023) and the open-source project GPT-J [4] (6000+ stars on GitHub since 2021) employ parallel-attention transformers, indicating that this architecture is indeed “quite popular”. Section 2 of the PALM paper already discussed findings on the efficiency of this architecture. We have cited both of these works in our paper and will further highlight these works in the related-works section. We apologize for any confusion caused by referring to this architecture as "parallel-attention transformers" in our paper instead of "parallel-layers" as used in the PALM paper. We will correct the naming of this architecture in the camera-ready version.
>
> Given the established findings from these popular works, it should not be expected that we need to reinvent their results through additional ablation experiments. Our focus remains on building upon these validated architectures to explore new avenues of research.
>
> [3] PaLM: Scaling Language Modeling with Pathways, Journal of Machine Learning Research 2023
>
>
> [4] GPT-J, 2021 https://github.com/kingoflolz/mesh-transformer-jax
>
>
> > Concern #3: Comparing the performance with BTX under the same inference throughput budget.
>
> Please see our global response for an analysis on inference efficiency both theoretically and empirically!
> TLDR:
> 1) BAM contains more FLOPs per token than the standard BTX. But under the optimal implementation, many of BAM’s FLOPs occurred due to attention experts can be “swept under the rug” by the parallel attention architecture we employ, as well as expert parallelization [5].
>
> 2) When we compare FLOPs between BAM and the parameter-matching variant of BTX, BAM contains approximately the same number of FLOPS while performing better model-quality wise (see Table 6 in the paper).
>
> 3) We also empirically ablate on inference latency, and show that our preliminary implementation of BAM inference is slightly slower than BTX. But as our paper mainly focuses on training gains rather than inference, our implementation is not optimal but can be greatly optimized for future work.
>
> [5] Switch Transformers: Scaling to Trillion Parameter Models with Simple and Efficient Sparsity
>
> > Concern #4:  [Authors should] understand the role of soft routing, instead of vanilla sparse routing, in the MoA layers either from empirical or theoretical perspectives.
>
>
> To address this concern, we have included additional ablation studies comparing soft routing with the most commonly used sparse routing approaches (i.e. top-1 and top-2 routing) in BAM’s attention experts layers. These experiments were conducted at the 590M scale, with all models given the same amount of compute. These results are in Table 3 of the PDF (see global rebuttal for PDF).  Our results indicate that BAM with top-1 and top-2 attention experts routing does not show improvement over the baseline BTX on most domains. Conversely, BAM with soft-routing consistently outperforms BTX across all domains.This demonstrates that using soft routing in MoA layers is crucial and supports our decision to implement this approach.
>
> We want to thank the reviewer again for their feedback. If you believe we have addressed your concerns adequately, we would greatly appreciate it if you could consider raising the score. Otherise, please let us know any remaining concerns so we can provide further details.

---

> > ### Comment · Reviewer_ScVn · 2024-08-10
> >
> > Thank the authors for their answers to my concerns. They have adequately answered every question I raised to my satisfaction, and therefore, I will increase my rating to 6.
> >
> > I wish this paper could be open-sourced, although it does not affect my scoring regarding NeurIPS policy. :)

---

### Official Review · Reviewer_MjaL · 2024-06-28

**Soundness:** 3
**Presentation:** 3
**Contribution:** 1
**Rating:** 5
**Confidence:** 4

**Summary:**

The paper proposes BAM (Branch-Attend-Mix), a novel approach to improve the training of Mixture of Experts (MoE) models by fully leveraging the parameters of pre-trained dense models. The authors introduce a method to initialize both feed-forward network (FFN) and attention layers from specialized dense models, enhancing the efficiency and performance of MoE training. The approach is evaluated on language models ranging from 590 million to 2 billion parameters, demonstrating superior performance in terms of perplexity and downstream task evaluations compared to existing methods.

**Strengths:**

1. The results demonstrate that BAM outperforms BTX under the same data and compute budget in both perplexity and downstream task evaluations.
2. The paper is exceptionally well-written, with a clear and logical flow that makes it easy to follow.

**Weaknesses:**

The technical novelty of this paper is relatively low, primarily because it combines existing methods, namely Branch-Train-Mix (BTX) and Mixture of Attention, without introducing any significant contributions. While the integration of these approaches is executed well, the paper does not offer significant new insights or methodologies beyond this combination.

**Questions:**

Why not include the baseline of standard upcycling?

**Limitations:**

Open-sourcing the implementation and models would significantly enhance the contribution of this paper.

---

> ### Author Rebuttal · Authors · 2024-08-07
>
> Dear reviewer,
>
> Thank you for your valuable feedback! We will recap your concerns/questions and address them one by one as follows:
>
> > Concern #1: Novelty of this work
>
> We respectfully disagree with the assessment that the combination of BTX and Mixture of Attention is trivial. We list some of the non-triviality below:
>
>
> 1) **Validation of Concept**: It cannot be simply assumed that combining Mixture of Attention with BTX would automatically yield superior results. Our hypothesis required ablation, which we undertook and presented in Section 7 of the paper. This testing confirmed that our approach does indeed enhance performance over simply scaling up the number of feed-forward experts.
>
> 2) **Challenges**: Naively combining the mixture of attention approach and BTX does not work. In fact, initial attempts using the typical top-1 and top-2 routing for the attention experts performed worse than the baseline BTX across most domains (See Table 3 in the PDF for these ablations). As a solution, we propose to use a soft-routing variant of the original Mixture of Attention that effectively addressed this.
>
> 3) **Efficiency Improvements**: To ensure that our model, BAM, operates efficiently, we adopted a parallel transformer architecture so that the attention experts and FFN experts can be computed in parallel. This hide away some of the additional computation costs incurred from adding the attention experts. Please see the global response for additional ablations on flops and latency for BAM. This adaptation was necessary to manage the increased complexity introduced by the combined methodologies.
>
> In the camera-ready version of our paper, we will further emphasize these non-trivial aspects of our research to clarify the contributions of our work. We urge the reviewer to reconsider the reject score due to “a lack of novelty”. We believe that a simple, effective, and “well-executed” solution should rather be applauded.
>
> > Question #2: Why not include the baseline of standard upcycling?
>
> We didn’t include it because it has already been established that BTX is a stronger baseline than sparse upcycling [1], and that we want to focus our limited compute on the main comparison and ablation showcasing the validity of our approach.
>
> > Concern #3: Not open-sourcing
>
> We understand the value of open-sourcing to the research community. However, our decision not to release the codebase and models is mainly due to the training framework and models are proprietary to the organization this work was done at. Despite these limitations, we have provided detailed descriptions of the hyperparameters, model architecture, and methods within our paper. This level of detail should enable readers to replicate our results using existing open-source LLM training codebases. Note that many significant works in the field, such as ones we build off of like PALM [2] and Switch Transformers [3] have also not been open-sourced but are still considered important contributions to the community. We are committed to contributing meaningfully to the field while navigating these complexities.
>
> [1] Branch-Train-MiX: Mixing Expert LLMs into a Mixture-of-Experts LLM, 2024
>
> [2] PaLM: Scaling Language Modeling with Pathways, JMLR, 2023
>
> [3] Switch Transformers: Scaling to Trillion Parameter Models with Simple and Efficient Sparsity, JMLR, 2022
> We want to thank the reviewer again for their feedback. If you believe we have addressed your concerns adequately, we would greatly appreciate it if you could consider raising the score. Otherise, please let us know any remaining concerns so we can provide further details.

---

> ### Comment · Reviewer_MjaL · 2024-08-08
>
> I agree the combination of BTX and Mixture of Attention is meaningful and I appreciate the author's efforts. However, if it's not open-sourced (I will raise my score to above 6 if it's open-sourced), I don't think this paper meets the bar of NeurIPS. I suggest the authors resubmit the paper to another venue.

---

> ### Author Response · Authors · 2024-08-08
> **Regarding Open-sourcing**
>
> Dear reviewer,
>
> Thank you for your feedback and the positive remarks.
>
> > I agree the combination of BTX and Mixture of Attention is meaningful and I appreciate the author's efforts
>
> We are pleased that our rebuttal has clarified the contributions of our work. We will ensure that the final version of the paper reflects the detailed explanations provided during the rebuttal phase.
>
> > However, if it's not open-sourced (I will raise my score to above 6 if it's open-sourced), I don't think this paper meets the bar of NeurIPS.
>
> We appreciate your support for open-sourcing, which we also firmly believe benefits the research community. However, due to intellectual property constraints, which are beyond the authors' control, we are unable to release the source code at this time. To aid reproducibility, we have meticulously detailed the model architecture and training setup in Appendix A and Section 5 of our paper, respectively.
>
> We also want to highlight that open-sourcing is not a "bar" for acceptance into NeurIPS main track.
> Many other LLM pre-training papers have been published at NeurIPS maintrack without open-sourcing such as Chinchilla [1], and the community has found their contributions to be important (1241 citations since 2022).
>
> Furthermore, the NeurIPS submission checklist explicitly state that the absence of open-source code should not be a ground for rejection. We quote the following directly from the NeurIPS checklist:
>
> >"**Question: Does the paper provide open access to the data and code, with sufficient instructions to faithfully reproduce the main experimental results, as described in supplemental material? While we encourage the release of code and data, we understand that this might not be possible, so “No” is an acceptable answer. Papers cannot be rejected simply for not including code…**"
>
> We hope this addresses your concerns, and we thank you for your consideration.
>
> [1] Training Compute-Optimal Large Language Models 2022, https://proceedings.neurips.cc/paper_files/paper/2022/file/c1e2faff6f588870935f114ebe04a3e5-Paper-Conference.pdf

---

> ### Author Response · Authors · 2024-08-10
> **NeurIPS Guidlines**
>
> Dear Reviewer,
>
> Thank you once again for your insightful comments and feedback on our work.
>
> As the rebuttal period ends in just three days, we wish to gently remind you that, according to NeurIPS reviewer guidelines, giving a paper a rejection score if it's not open-sourced is against the rules. The NeurIPS submission checklist clearly states: **"While we encourage the release of code and data, we understand that this might not always be possible. Papers cannot be rejected simply for not including code."**
>
> We kindly ask that you follow these guidelines and consider adjusting the score to "above 6" you previously committed to.
>
> Thank you for your consideration,
>
> Authors of Submission 19862

---

> ### Comment · Reviewer_MjaL · 2024-08-11
>
> Dear authors,
> Thank you for your rebuttal and reminder of the policy. I'd like to increase my score to 5.

---

> ### Author Response · Authors · 2024-08-11
> **Response to Reviewer Comments**
>
> Dear reviewer,
>
> Thank you for engaging with our rebuttal.
>
> We would appreciate it if you could clarify any lingering valid concerns you have regarding our work, which might justify your rating of 5. We note that the NeurIPS reviewer guide advises using a borderline score of 5 sparingly.
>
> We believe that the inability to open-source our code (due to constraints beyond our control) cannot justify a low score like 5. Particularly since 1) As mentioned above, NeurIPS reviewer guidelines does not hold open-sourcing as a requirement 2) Our paper is neither a dataset nor a benchmark study whose primary contribution is the code.
>
> In case you might have missed it, we have conducted two additional ablation studies plus updated our existing results with more appropriate benchmark tasks during the rebuttal phase. These improvements are detailed in our response to Reviewer ScVn.  In addition, we have incorporated your feedback to enhance the clarity significance of our work’s contributions in the forthcoming version of the paper. We believe these efforts have significantly strengthened the paper quality.
>
> With these considerations, we urge you to reconsider and honor your previous indication of “raise the score to above 6”. If not, please let us know your remaining valid concerns.
>
> Thank you for your time,
>
> Authors of 19862

---

> > ### Author Response · Authors · 2024-08-12
> >
> > Dear reviewer,
> >
> > As the rebuttal period ends in just one day, we wish to gently request your feedback above once more.
> >
> > We would appreciate it if you could share any remaining valid concerns you have regarding our work to justify your rating of 5, which has been advised to be used sparingly. We would be happy to try to address any remaining concerns you have.
> >
> > Thank you again for all your time and feedback on our work,
> >
> > Authors of 19862

---

### Official Review · Reviewer_4yUX · 2024-07-15

**Soundness:** 4
**Presentation:** 3
**Contribution:** 2
**Rating:** 6
**Confidence:** 4

**Summary:**

The paper extends previous work (BTX), which combines different expert LLMs into an MoE model by a) building FFN experts,  b) averaging the remaining parameters, and c) training a router over the experts. The authors propose BAM, which uses Mixture of Attention (MoA) to consolidate the different attention modules across experts, rather than parameter averaging.

The authors make some modifications on the underlying architecture as well, using soft routing across all attention experts, and employing parallel FFN / Attention blocks to absorb the cost of this more expensive attention layer. They also propose
Experiments are conducted on both small and large-scale base LLMs, using 3 expert domains (code, law, and math).

The authors demonstrate that overall, BAM outperforms the BTX baselines over both data and training compute budgets, for both in-distribution perplexity and common benchmarks.

**Strengths:**

1. The paper proposes a single fix (mixture-of-attention) to address the limitations of parameter averaging in previous work
2.  Performance gains are somewhat small, but consistent across scale and benchmarks
3.  The authors do a good job at taking into account both training data and training compute into their analysis to properly evaluate their method.

**Weaknesses:**

1. It's unclear to me what the additional cost (both in flops and in latency) that is incurred by having to go through all Attention Experts. This raises questions on the scalability of the method to more expert domains. Can the authors compare FLOP / latency at inference time for both BAM and BTX ?

**Questions:**

1. I am not sure I fully understood the different BAM attentions. BAM with KV Sharing is essentially the standard MoA implementation, akin to multi-query attention. For BAM with KV experts, you run the full attention module, and perform a weighted average of the individual attention outputs ?
2. Could you expand on why you add 10% of text from Common Crawl in every domain ?
3. For the first ablation, how exactly did you add new experts ? Did you add new domains, or did you split domain(s) into multiple experts ?
4. For the second ablation, was the model trained with top-3 ? What is the performance of top-1 model at that scale ?

**Limitations:**

Yes

---

> ### Author Rebuttal · Authors · 2024-08-07
>
> Dear reviewer,
>
>
> Thank you for your valuable feedback! We will recap your concerns/questions and address them one by one as follows
>
> > Question 1: Can the authors compare FLOP / latency at inference time for both BAM and BTX
>
> Please see our global response for an analysis on inference efficiency. TLDR:
>
> 1) BAM contains more FLOPs per token than the standard BTX. But under the optimal implementation, many of BAM’s FLOPs occurred due to attention experts can be “swept under the rug” by the parallel attention architecture we employ, as well as expert parallelization [1].
>
>
> 2) When we compare FLOPs between BAM and the parameter-matching variant of BTX, BAM contains approximately the same number of FLOPS while performing better model-quality wise (see Table 6 in the paper).
>
>
> 3) We also empirically ablate on inference latency, and show that our preliminary implementation of BAM inference is slower than BTX. But as our paper mainly focuses on training gains rather than inference, our implementation is not optimal but can be greatly optimized for future work.
>
>
> [1] Switch Transformers: Scaling to Trillion Parameter Models with Simple and Efficient Sparsity
>
>
> > Question 2: Clarifications on BAM’s KV sharing architecture
>
> For BAM with KV Experts, each expert runs its full attention module independently. Afterward, we perform a weighted average of the outputs from these attention experts to produce the final output.
>
> BAM with KV Sharing operates similarly to the BAM with KV experts, but with a key distinction: the key and value parameters are shared among all experts to enable a single kv computation for the whole MoA layer which leads to decrease in compute and memory requirement. The aggregation remains the same, where a weighted average of the individual attention experts is used to obtain the final output.
>
> We acknowledge the need for clearer explanations of these mechanisms and will provide further details in the camera-ready version of the paper!
>
> > Question 3: Why you add 10% of text from Common Crawl in every domain?
>
> In our ablation experiments, we observed that incorporating a portion of general-text data into the training mix enhances the expert dense model’s performance since it allows a portion of the data matching with the data distribution of the seed model. We specifically chose Common Crawl because it is one of the largest genera-text datasets we have available, which allows us to minimize seeing repeated tokens during training. In future work, one can further explore what is the most optimal data mixture to use.
>
> > Question 4: For the first ablation, how exactly did you add new experts ? Did you add new domains, or did you split domain(s) into multiple experts.
>
> All new experts are initialized from the same seed dense model. We do this because we assume we are working with a fixed set of existing seed and expert dense models.
>
> > Question 5: For the second ablation, was the model trained with top-3 ? What is the performance of top-1 model at that scale.
>
>
> In the second ablation study, BAM uses soft-routing for 4 attention experts and top-1 routing for 4 FFN experts. In contrast, BTX uses top-3 routing for 6 FFN experts. We selected this exact setup for BTX so that it matches the number of total parameters as well as active parameters with the BAM experiments shown in Tables 2 and 3 in the paper. These two runs were token matched, meaning they were trained on the exact same training data.
>
> We did not test BTX with top-1 routing for 6 FFN experts. This decision was based on the observation that, in Mixture of Experts, performance typically improves as the 'k' value increases in top-k routing due to the increased number of active parameters. Therefore, we anticipated that top-1 routing would underperform compared to top-3 under these conditions.
>
>
> We want to thank the reviewer again for their feedback. If you believe we have addressed your concerns adequately, we would greatly appreciate it if you could consider raising the score. Otherise, please let us know any remaining concerns and we will provide further details.

---

> > ### Comment · Reviewer_4yUX · 2024-08-12
> > **Re: rebuttal**
> >
> > Thank you for addressing most of my concerns, and for clarifying key points.
> >
> > Regarding my second to last point, I apologize if my question was not clear. Given that each added experts is copied from the base seed model and then finetuned on its own (unique?) data mixture, how exactly did you train 8 experts if there are only 4 data domains ?
> >
> > Overall I am happy with the clarifications provided by the authors.

---

> > > ### Author Response · Authors · 2024-08-12
> > > **Reply**
> > >
> > > Dear reviewer,
> > >
> > > Thank you for your feedback and questions.
> > >
> > > > Given that each added experts is copied from the base seed model and then finetuned on its own (unique?) data mixture, how exactly did you train 8 experts if there are only 4 data domains?
> > > Here's the 3 stages for obtaining this compute-matched variant of BTX:
> > > 1) We start with the dense seed model and create three copies  (this step is identical to BAM)
> > > 2) Each of these three models is independently trained on its specific data domains. The three domains were law, math, and code. (This step is also identical to BAM)
> > > 3) We initialized three of the MoE’s FFN experts using the three expert dense models. The remaining MoE experts are all initialized from the same FFN from the seed dense model. This approach is similar to [1], where if the number of different dense models available is less than the number of experts, the same dense model's FFN is replicated multiple times to upcycle into multiple experts. Following this initialization, the entire MoE model is trained (including the expert parameters) for a small number of steps, just like BAM.
> > >
> > > We hope this clarifies the question, and please let us know if anything remains unclear. We will also add these clarification details to the next version of the paper.
> > >
> > > In addition, we have conducted two additional ablation studies plus updated our existing results with more appropriate benchmark tasks during the rebuttal phase. These improvements are detailed in our response to Reviewer ScVn, and we believe these efforts have significantly strengthened the paper quality. If you're happy with our rebuttals, the authors would really appreciate if you might consider raising the score! :)
> > >
> > > Thank you for your time,
> > >
> > > Authors of Submission19862
> > >
> > > [1]  Sparse Upcycling: Training Mixture-of-Experts from Dense Checkpoints, 2022

---

> > > > ### Comment · Reviewer_4yUX · 2024-08-12
> > > > **Re: rebuttal**
> > > >
> > > > Thank you for the clarification.

---

### Author Rebuttal · Authors · 2024-08-07

Dear reviewers,

Thank you again for your valuable feedback!

In the attached PDF, we have provided updated experiments and additional ablations.

In addition, see blow for additional analysis on inference efficiency.

# Inference Arithmetic

We analyze the parameter and arithmetic for BTX vs. BAM during inference (the forward pass). Table 1 below shows the estimates for "FLOPs per Token" per Transformer layer. We use the common FLOPs counting methodology employed [1] and [2], where we exclude non-linearities, biases, and layer normalization which are negligible.

**Table 1**
| **Operation**                 |           | **BTX**               | **BAM**                        |
|-------------------------------|-----------|-----------------------|--------------------------------|
| **Attention Router**          | Params    | -                     | $n_{experts}d_{model}$         |
|                               | FLOPs     | -                     | $2n_{experts}d_{model}$        |
| **Attention: QKV**            | Params    | $d_{model}3d_{attn}$  | $n_{experts}d_{model}3d_{attn}$|
|                               | FLOPs     | $6d_{model}^2$        | $6n_{experts}d_{model}^2$      |
| **Attention: Mask**           | Params    | -                     | -                              |
|                               | FLOPs     | $2n_{ctx}d_{model}$   | $2n_{experts}n_{ctx}d_{model}$ |
| **Attention: Projection**     | Params    | $d_{attn}d_{model}$   | $n_{experts}d_{attn}d_{model}$ |
|                               | FLOPs     | $2d_{model}^2$        | $2n_{experts}d_{model}^2$      |
| **FFN Router**                | Params    | $n_{experts}d_{model}$| $n_{experts}d_{model}$         |
|                               | FLOPs     | $2n_{experts}d_{model}$| $2n_{experts}d_{model}$       |
| **FFN***                      | Params    | $1.5n_{experts}d_{model}d_{ff}$ | $1.5n_{experts}d_{model}d_{ff}$|
|                               | FLOPs     | $3n_{topk}d_{model}d_{ff}$ | $3n_{topk}d_{model}d_{ff}$|





# Theoretical and Empirical Anlysis
## Theoretical FLOPS Counting

Using the arithmetic formulation above, Table 2 examines the inference FLOPs on the small-scale experiments (see Appendix A). Row 1 shows BAM with KV experts, row 2 shows the standard BTX, and row 3 shows a parameter-matching variant of BTX with 6 FFN experts & top-3 routing (refer to Table 6 in the paper). We use a prompt with context length of $256$.

**Table 2**
| Method                  | **$n_{experts}$** | **$n_{topk}$** | Total Param | Attention FLOPs | FFN FLOPs  | Total FLOPs |
| ----------------------- | ------------- | ---------- | ----------- | --------------- | ---------- | ----------- |
| BAM                     | 4             | 1          | 776M        | 35,651,584      | 12,589,056 | 48,257,024  |
| BTX (Standard)| 4             | 1          | 700M        | 8,912,896       | 12,589,056 | 21,510,144  |
| BTX (Param Matching)    | 6             | 3          | 776M        | 8,912,896       | 37,767,168 | 46,692,352  |



## BAM Efficiency Optimization
Compared to BTX (standard), BAM consumes more FLOP due to the soft-routing attention experts. To address the increase in FLOPs, we implement the following optimizations:
1) All attention experts are computed in parallel via expert parallelism [3].
2) The attention experts and FFN experts are computed in parallel rather than sequentially (see Figure 1 in the paper), using this parallel reformulation from [4]: $$y = x + MLP(LayerNorm(x)) \textbf{+} Attention(LayerNorm(x))$$

In Table 2, row 3, we see that each attention expert is significantly less FLOP intensive than each FFN expert. Thus, with the parallel transformer block, we are computing the FLOP-expensive and FFN expert(s) alongside the less FLOP-expensive and smaller attention expert(s), all while using expert parallelism and effectively overlapping computations. This way we are effectively overlapping compute.

## Empirical Inference Latency
For empirical inference latency, we generate 16 tokens given a prompt with $256$ length, averaged over 10 runs on TPU v4s; we note BAM slightly is slower than BTX:
    BAM with KV experts: `6.17s`
    BTX (standard): `4.81s`.

Although we overlap computation with the parallel transformers architecture and expert parallelism, BAM's longer inference time is likely casued by increased memory pressure from attention experts (each expert has a KV cache) as attention is memory-bound [5]. Motivated by this, the KV-sharing version of BAM reduces the inference latency from `6.17s` to `5.96s`, as all attention experts share one KV cache. We believe we can further optimize inference time to be close to BTX baseline (representative of standard MoEs) as our hardware and framework code is optimized for training rather than inference latency.



## TLDR
Takeaways:
1) BAM contains more FLOPs than BTX, but under optimal implementation, many of BAM’s FLOPs due to attention experts can be "swept under the rug" by the parallel attention architecture and expert parallelism.
2) When we compare FLOPs between BAM and the parameter-matching variant of BTX, BAM contains approximately the same number of FLOPS while performing better model-quality wise (see Table 6 in the paper).
3) We also empirically ablate on inference latency, and show that our preliminary inference implementation of BAM is slower than BTX. But as our paper mainly focuses on training gains rather than inference, our primary implementation is not optimal but can be greatly optimized for future work.



# Reference
[1] Scaling Laws for Neural Language Models, 2020

[2] SEER-MoE: Sparse Expert Efficiency through Regularization for Mixture-of-Experts, 2024.

[3] Switch Transformers: Scaling to Trillion Parameter Models with Simple and Efficient Sparsity, 2021

[4] PaLM: Scaling Language Modeling with Pathways, 2022.

[5] Full Stack Optimization of Transformer Inference: a Survey, 2023.

---

### Decision · Program_Chairs · 2024-09-25

**Decision:**

Accept (poster)

**Comment:**

The paper looks into methodologies for training high quality MoE models. In particular, the paper proposes BAM, a simple yet effective parameter upcycling method, which extends prior work Branch-Train-Mix (BTX) while also leveraging Mixture of Attention (MoA) to consolidate attention modules across experts. The paper also explores additional design choices, such as soft routing and parallel attention. Overall, evaluation shows promising results on small to large scale LLMs across different domains.

How to train high quality MoE is a timely topic, and the paper has made some interesting observations, which shall help the community improve the training of MoEs. The writing of the paper is also of high quality, and the reviews are overall positive. Therefore, I recommend acceptance to this paper.